# Long-Term Protection Elicited by an Inactivated Vaccine Supplemented with a Water-Based Adjuvant against Infectious Hematopoietic Necrosis Virus in Rainbow Trout (*Oncorhynchus mykiss*)

Yujie Lin,[a,b,c] Guangming Ren,[a,b] Jingzhuang Zhao,[a,b] Yizhi Shao,[a,b] Baoquan He,[a,b] Xin Tang,[a,b] Ou Sha,[a,b] Wenwen Zhao,[a,b,c] Qi Liu,[a,b] (ID) Liming Xu,[a,b] Tongyan Lu[a,b]

aDepartment of Aquatic Animal Diseases and Control, Heilongjiang River Fisheries Research Institute, Chinese Academy of Fishery Sciences, Harbin, China
bKey Laboratory of Aquatic Animal Diseases and Immune Technology of Heilongjiang Province, Harbin, China
cUniversity of Dalian Ocean University, College of Fisheries and Life Sciences, Dalian, China

Liming Xu and Tongyan Lu contributed equally to this study.

**ABSTRACT** Previous inactivated vaccines against infectious hematopoietic necrosis (IHN) usually had a strong early immune protective effect but failed to provide long-term protection in rainbow trout (*Oncorhynchus mykiss*). To find a method for stabilizing the desired protective effect of IHN vaccines, we assessed the immune enhancement effect of four adjuvants on formaldehyde inactivated vaccine for IHN at 60 days postvaccination (dpv). The efficacy of a two-dose vaccination with the candidate adjuvant-formaldehyde inactivated vaccine for IHN was evaluated in terms of early protection and long-term protection (30 to 285 dpv). Neutralizing antibody titers were also measured at each time point. The Montanide GEL 02 PR (Gel 02) adjuvant significantly enhanced the immune protection provided by the IHN inactivated vaccine, whereas the immune-boosting effect of the other tested adjuvants lacked statistical significance. Both tested Gel 02-adjuvanted IHN inactivated vaccine dosages had a strong immune protection effect within 2 months postvaccination, with a relative percent of survival (RPS) of 89.01% to 100%, and the higher dosage provided complete protection at 204 dpv and a RPS of 60.79% on 285 dpv by reducing viral titers in rainbow trout. The neutralizing antibodies were observed only in vaccinated fish on 30 and 60 dpv. Through compatibility with an appropriate adjuvant, the highly immune protective effect of an IHN inactivated vaccine was prolonged from 60 dpv to at least 284 dpv; this novel adjuvant-IHN inactivated vaccine has promise as a commercial vaccine that provides the best available and longest duration of protection against IHN to rainbow trout.

**IMPORTANCE** Infectious hematopoietic necrosis virus (IHNV) is one of the most serious pathogens threatening the global salmon and trout industry. However, there is currently only one commercialized infectious hematopoietic necrosis (IHN) vaccine, and it is inadequate for solving the global IHN problem. In this study, a promising adjuvanted inactivated vaccine with long-term protection was developed and comprehensively studied. We confirmed the presence of a late antiviral response stage in vaccinated rainbow trout that lacked detectable neutralizing antibodies, which are commonly recognized to be responsible for long-term specific protection in mammals. These findings further our understanding of unique features of fish immune systems and could lead to improved prevention and control of fish diseases.

**KEYWORDS** adjuvant, inactivated vaccine, infectious hematopoietic necrosis virus, long-term protection, rainbow trout

**Ad Hoc Peer Reviewer** (ID) Lingbing Zeng, Yangtze River Fisheries Research Institute; Weiwei Zeng, Pearl River Fisheries Research Institute

Address correspondence to Liming Xu, xuliming@hrfri.ac.cn, or Tongyan Lu, lutongyan@hrfri.ac.cn.

The authors declare no conflict of interest.

**S**almon and trout aquaculture represents one of the most active industries in fisheries. According to the relevant data from the United Nations Food and Aquaculture Organization (FAO) (1), salmon and trout have been the most important traded commodity in value terms since 2013, and they accounted for approximately 18% of the total value of internationally traded fish products in 2018. However, the global salmon and trout farming industry is under constant threat from infectious hematopoietic necrosis (IHN). IHN, caused by infectious hematopoietic necrosis virus (IHNV), is an acute infectious disease of many salmon and trout species (2, 3). In some cases, epidemics of IHN can result in losses of >90%, depending on factors such as the fish species, virus strain, and environmental conditions (4). The World Organization for Animal Health (OIE) lists IHN as a notifiable animal disease (5). IHN was first reported in the 1950s in hatchery-reared sockeye salmon (*Oncorhynchus nerka*) fry in Washington and Oregon, USA (6). IHNV has now spread to many countries in Asia, Europe, and Africa, causing a heavy burden on global seawater and freshwater aquaculture of salmonids (2).

Considerable efforts have been made over the past several decades to develop vaccines against IHN, and the main studied vaccine types include inactivated vaccines (7 to 10), attenuated live vaccines (11, 12), subunit vaccines (13), and DNA vaccines (14, 15). In 2005, an IHN DNA vaccine (Apex-IHN) (16) was approved in Canada, becoming the first and only IHN vaccine to be successfully marketed. However, one commercial vaccine alone cannot effectively prevent the global epidemic of IHN. To solve the global problem caused by endemic IHN, more vaccines need to be developed and commercialized in more countries.

The DNA vaccines for IHN were proven to provide adequate long-term (~2 years) protection against IHNV to rainbow trout (3, 4, 14). However, because DNA vaccines are defined as transgenic products, which require extra safety assessments, they have a much longer commercialization period compared with non-genetically modified organism (GMO) vaccines. In contrast, inactivated vaccines are commonly recognized as highly safe and stable; consequently, such vaccines have become dominant among approved commercial aquatic vaccines (17). Candidate inactivated vaccines against IHNV that have promising protection efficacy in rainbow trout have been developed (7 to 9). These inactivated vaccines induce extremely rapid innate immune responses and early specific immune responses. However, the immune protective effect of these IHN inactivated vaccines was determined mainly within approximately 2 months after immunization, and their long-term protection in rainbow trout has not been evaluated. In a previous study (10), we found that the protective efficacy of an IHN inactivated vaccine was significantly high at earlier time points but became insufficient after 2 months postimmunization. Inadequate long-term protection is a common problem of inactivated vaccines. To stabilize their protective effect, inactivated vaccines are often used in conjunction with appropriate adjuvants (18), and adjuvants have been applied to the inactivated vaccines developed for many kinds of fish, such as coho salmon (*Oncorhynchus kisutch*) (19), turbot (*Scophthalmus maximus*) (20), olive flounder (*Paralichthys olivaceus*) (21), Atlantic salmon (*Salmo salar*) (22), and rainbow trout (23), which clearly illustrates that the adjuvants can provide satisfactory enhancement of the immune protection induced by vaccines. However, there have been very few studies on adjuvants in IHN inactivated vaccines, and their evaluations focused only on early protection (7, 9).

The goal of the present study was to assess whether the use of an appropriate adjuvant could allow IHN inactivated vaccines to induce long-term immune protection in rainbow trout. We screened a variety of adjuvants to identify the optimal one suitable for use with the formaldehyde-inactivated vaccine against IHN, and the long-term protection effect of the adjuvanted inactivated vaccine in rainbow trout was comprehensively analyzed.

## RESULTS

**Screening of suitable IHNV inactivators.** The protection efficacy of the IHN inactivated vaccines prepared with formaldehyde or beta-propiolactone (BPL) was compared by challenging two sizes of vaccinated rainbow trout (average weight of $6 \pm 2$ g

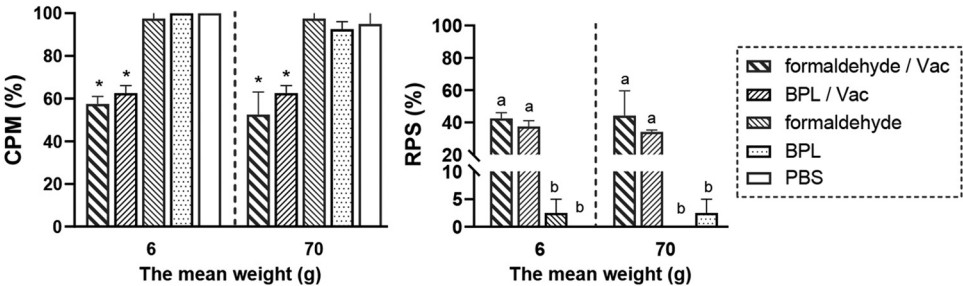

**FIG 1** The protective effect of the IHN inactivated vaccines prepared with formaldehyde or BPL in rainbow trout determined at 60 dpv. Two groups of rainbow trout, one with an average weight of 6 ± 2 g and the other with an average weight of 70 ± 8 g, were vaccinated with the IHN inactivated vaccines at dosages of 50 µL or 100 µL, respectively, and then were challenged with IHNV strain LN15 (100 $TCID_{50}$) at dosages of 50 µL or 100 µL, respectively. *, $P < 0.05$ versus the PBS control group. Different letters at the top of a bar indicate significant differences ($P < 0.05$).

and 70 ± 8 g) at 60 dpv. The results show that both of the vaccines could provide protection to both sizes of rainbow trout, and the cumulative percentage mortality (CPM) of fish in the vaccination group was significantly lower ($P < 0.05$) than that in the inactivator alone or phosphate-buffered saline (PBS) groups, in which the CPM was higher than 90%. The relative percents of survival (RPS)s of the formaldehyde-inactivated vaccine against IHN in the two sizes of rainbow trout were 42.5% (fish weighing 6 ± 2 g) and 44.15% (fish weighing 70 ± 8 g) (Fig. 1); these values both trended higher compared with those of the BPL-inactivated vaccine against IHN, but these differences failed to reach statistical significance (Fig. 1; $P > 0.05$). Owing to its low cost and wide application in commercial vaccines, formaldehyde was selected for use in preparing the IHN inactivated vaccines in this study.

**The protection effect of IHN inactivated vaccines supplemented with different adjuvants. (i) Immune protection efficacy of adjuvanted inactivated vaccines against IHN.** Triple replicate subgroups of 20 rainbow trout (weighing 7 ± 1.5 g) that had been vaccinated with an IHN inactivated vaccine or an adjuvant alone or PBS (negative control) were challenged at 60 dpv. The prepared Montanide ISA 763A VG-adjuvanted vaccine, which had excessive viscosity, was found to be difficult to inject intraperitoneally, so it was excluded from this study. The results showed that the rainbow trout immunized with any of the adjuvanted IHN inactivated vaccines were significantly protected compared with those treated with the adjuvants alone or mock-immunized with PBS ($P < 0.05$), and fish in the vaccination groups exhibited CPMs of 15% to 65% compared with the fish treated with adjuvants or mock-immunized with PBS, which exhibited CPMs of 80% to 85% (Fig. 2). The RPSs of the groups vaccinated with the non-adjuvanted vaccine or the vaccines supplemented with aluminum hydroxide gel, Montanide GEL 02 PR, or GR208 were 30.33%, 30.51%, 78.68%, and 36.4%, respectively. The highest RPS was observed in the Montanide GEL 02 PR-adjuvanted vaccine group, and this value was significantly different from those of the other groups ($P < 0.05$). However, no significant difference was observed among the aluminum hydroxide gel-adjuvanted vaccine group, the GR208-adjuvanted vaccine group, and the non-adjuvanted vaccine group ($P > 0.05$). These results indicate that the IHN inactivated vaccine in combination with the Montanide GEL 02 PR adjuvant had a higher protective effect compared with the non-adjuvanted vaccine against IHNV challenge, whereas the addition of aluminum hydroxide gel or GR208 as adjuvants to the IHN inactivated vaccine did not significantly enhance the immune protection induced by the vaccine. Thus, Montanide GEL 02 PR was selected as the optimal adjuvant to use in the preparation of IHN inactivated vaccine.

**(ii) NAb titers in fish vaccinated with IHN inactivated vaccines supplemented with different adjuvants.** At 60 dpv, the blood of treated rainbow trout ($n = 5$) was collected from their caudal veins to determine the NAb titers. The results show that the mean NAb titers from the vaccine groups were all higher than 20, whereas the titers in the vaccine-free groups were all lower than 20; specifically, the mean NAb

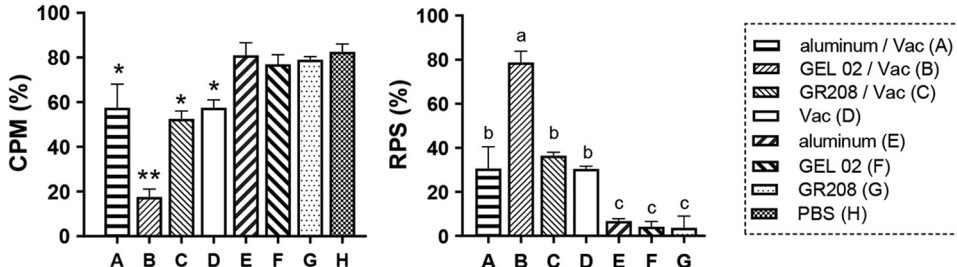

**FIG 2** Protective effect of the IHN inactivated vaccine alone or in combination with different adjuvants. Rainbow trout (weighing $7 \pm 1.5$ g) were vaccinated and challenged with IHNV strain LN15 (100 TCID$_{50}$) at a dosage of 50 $\mu$L at 60 dpv. The prepared Montanide ISA 763A VG-adjuvanted vaccine was difficult to inject, so it was not included in these experiments. *, $P < 0.05$ versus the PBS control group. **, $P < 0.01$ versus the PBS control group. Different letters at the top of a bar indicate significant differences ($P < 0.05$).

titers in the aluminum hydroxide gel-adjuvanted, Montanide GEL 02 PR-adjuvanted, GR208-adjuvanted, and nonadjuvanted vaccine groups were 35.61, 62.12, 32.50, and 30.85, respectively (Table 1). The NAb titer in the Montanide GEL 02 PR-adjuvanted vaccine group was significantly higher than that in any other vaccine group ($P < 0.05$), but no significant differences were observed among the aluminum hydroxide gel-adjuvanted, GR208-adjuvanted, or non-adjuvanted vaccine groups ($P > 0.05$). The results are consistent with those of the immunity protection efficacy tests.

**The optimal vaccine dosage of the Montanide GEL 02 PR-adjuvanted IHN inactivated vaccine.** Triple replicate groups of 15 rainbow trout (weighing $6 \pm 2$ g) that had been immunized with different dosages of Montanide GEL 02 PR-adjuvanted IHN inactivated vaccine or mock-vaccinated with PBS were challenged with 50 $\mu$L of IHNV strain LN15 (100 TCID$_{50}$) at 30 dpv. The rainbow trout in each vaccinated group were significantly better protected compared with those in the PBS-mock-vaccinated group ($P < 0.05$); the vaccine groups had average CPMs of 0% to 12.5%, whereas the average CPM in the PBS-mock-vaccinated control group was 81.25% (Fig. 3). The RPSs in Montanide GEL 02 PR-adjuvanted vaccine groups that contained 9, 45, or 90 $\mu$L of inactivated IHNV suspension were not significantly different from one another (88%, 100%, or 85.5%, respectively; $P > 0.05$) (Fig. 3). These results show that the Montanide GEL 02 PR-adjuvanted IHN inactivated vaccine had a significant protective effect on rainbow trout at all tested dosages.

**Duration of immune protection. (i) Duration of immune protection induced by Montanide GEL 02 PR-adjuvanted IHN inactivated vaccine (containing 9 $\mu$L of inactivated IHNV).** Triple replicate groups of 20 rainbow trout (weighing $8 \pm 2.5$ g) that had been intraperitoneally injected with 50 $\mu$L of Montanide GEL 02 PR-adjuvanted vaccine (containing 9 $\mu$L of inactivated IHNV) or PBS were challenged with IHNV strain LN15 at 28, 56, 70, 144, or 223 dpv. At the earliest three time points, fish in the immunization group were significantly protected compared with those in the PBS-mock-vaccinated group ($P < 0.05$), and the average CPM in the immunization group ranged from 0% to 15%, whereas the average CPM in the PBS control group was 70% to 85% (Fig. 4). At 144

**TABLE 1** Neutralizing antibody (NAb) elicited by different inactivated vaccines on 60 dpv

| Groups | Treatments | The mean NAb titers ($n = 5$)[a] |
|---|---|---|
| 1 | Aluminum hydroxide gel vaccine | $35.61 \pm 8.67$ |
| 2 | Montanide GEL 02 PR vaccine | $62.12 \pm 11.35$* |
| 3 | GR208 vaccine | $32.50 \pm 5.62$ |
| 4 | Adjuvant-free vaccine | $30.85 \pm 6.51$ |
| 5 | Aluminum hydroxide gel | $<20$ |
| 6 | Montanide GEL 02 PR | $<20$ |
| 7 | GR208 | $<20$ |
| 8 | PBS control | $<20$ |

[a]*, $P < 0.05$ versus other groups.

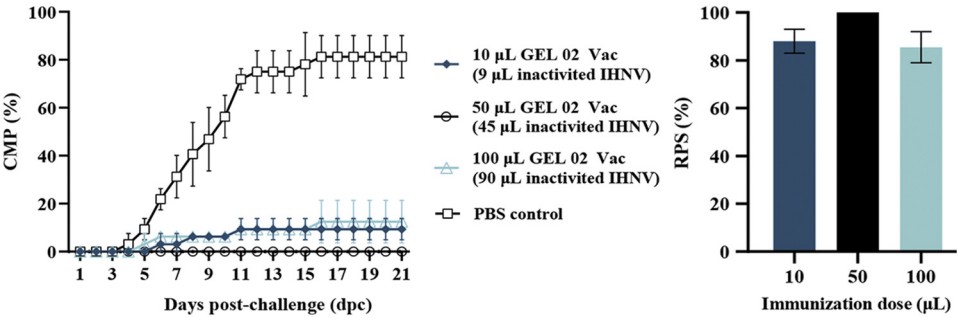

**FIG 3** Protective effects elicited by different dosages of Montanide GEL 02 PR-adjuvanted IHN inactivated vaccine at 30 dpv. Fish mock-vaccinated with PBS were used as the negative control. No significant differences in mortality were observed between any replicate within each treatment group.

and 223 dpv, the CPMs from vaccine and PBS control groups did not differ significantly; their mean CPMs were 65% and 73.35%, respectively ($P > 0.05$). At each postvaccination challenge time point, the fish in the PBS control group generally began dying at 3 to 6 days postchallenge (dpc), and deaths continued occurring for a total period of approximately 12 to 15 days, whereas in all the vaccine groups except those challenged at 144 or 223 dpv, the onset of deaths was delayed and the total period of deaths lasted only a few days.

**(ii) Duration of immune protection of Montanide GEL 02 PR-adjuvanted IHN vaccine (containing 45 μL of inactivated IHNV).** Triple replicate groups of 20 rainbow trout (weighing 8 ± 2 g) that had been intraperitoneally injected with 50 μL of Montanide GEL 02 PR-adjuvanted IHN inactivated vaccine (containing 45 μL of inactivated IHNV) or PBS were challenged with IHNV strain LN15 at 30, 60, 188, 204, or 285 dpv. The average CPMs in the vaccine groups ranged from 0% to 32%, and those in the PBS control groups ranged from 67.5% to 92.5%, which were significantly higher than those in the vaccine groups (Fig. 5; $P < 0.05$). The RPSs at the time points of 30, 60, 188, and 204 dpv ranged from 89.01% to 100% and were all significantly higher ($P < 0.05$) than the RPS at 285 dpv (60.79%), but no significant difference in RPS was observed among the first four time

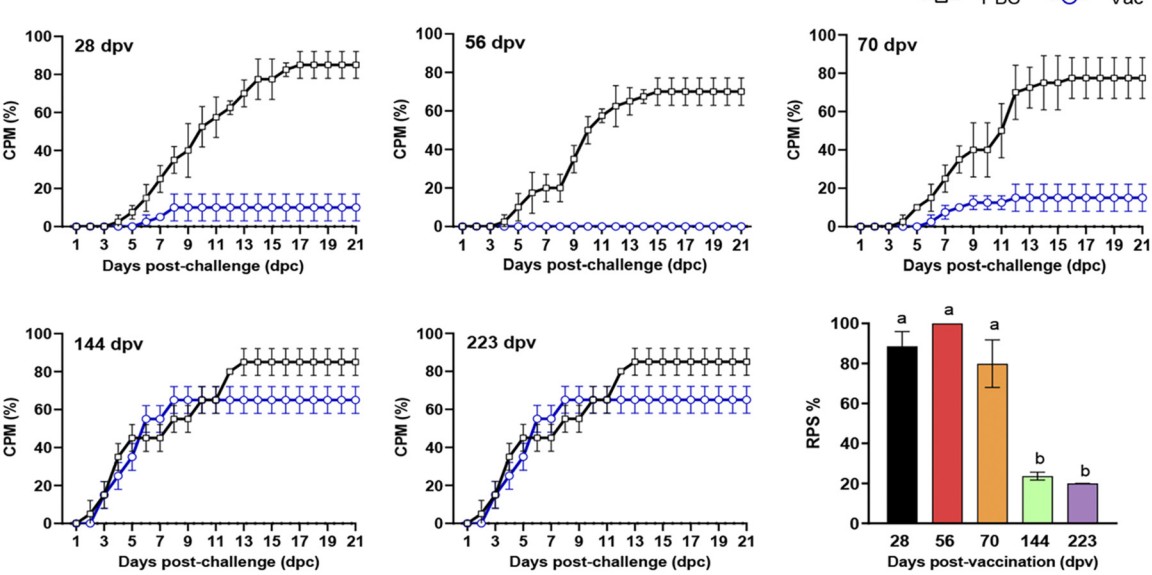

**FIG 4** Protective effect of Montanide GEL 02 PR-adjuvanted IHN inactivated vaccine (containing 9 μL of inactivated IHNV) in rainbow trout at different postvaccination time points. The immunized fish were challenged with specific dosages of IHNV strain LN15 at different postvaccination time points (Table 5). No significant differences in mortality were observed between any replicates within each treatment. Different letters on the top of a bar indicate significant differences ($P < 0.05$).

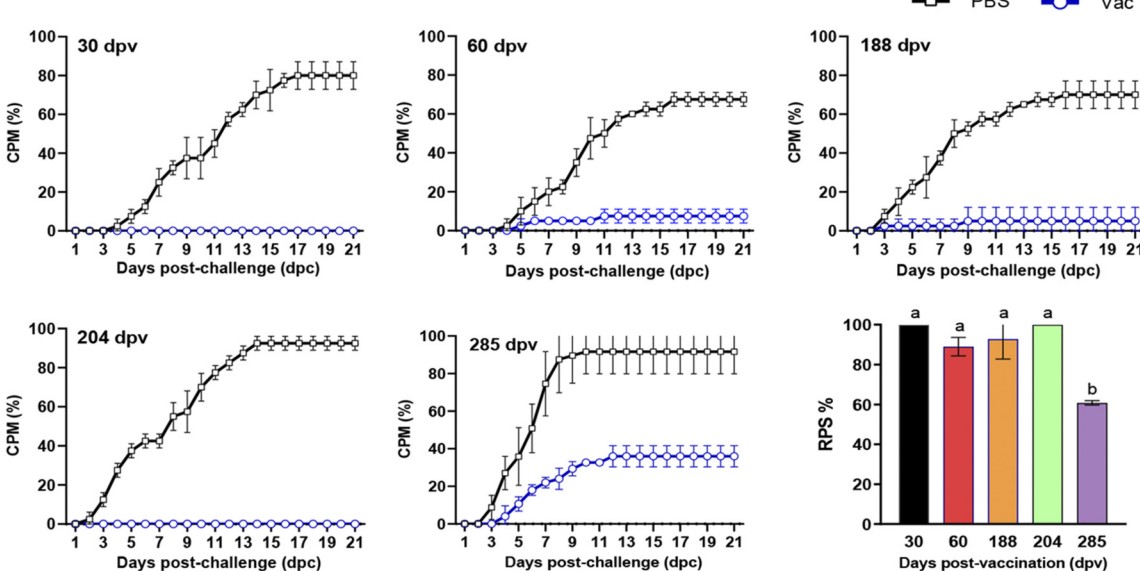

**FIG 5** Protective effect of Montanide GEL 02 PR-adjuvanted IHN inactivated vaccine (containing 45 $\mu$L of inactivated IHNV) in rainbow trout at different postvaccination time points. The immunized fish were challenged with specific dosages of IHNV strain at different postvaccination time points (Table 5). No significant differences in mortality were observed between any replicates within each treatment group. Different letters at the top of a bar indicate significant differences ($P < 0.05$).

points ($P > 0.05$). These results indicate that the immune protective effect of a 50-$\mu$L dosage (containing 45 $\mu$L of inactivated IHNV) of Montanide GEL 02 PR-adjuvanted IHN inactivated vaccine in rainbow trout could last for at least 285 days (9.5 months), and extremely high protection effects could be maintained for at least 204 days.

**(iii) Duration of NAb elicited by Montanide GEL 02 PR-adjuvanted IHN inactivated vaccine.** At 30, 60, 188, 204, and 285 dpv, serum samples from unchallenged fish ($n = 5$) were tested for seroconversion using a NAb assay. A NAb titers of <20 was considered negative. The results show that the NAb titer of the vaccine group increased first and then decreased, peaking at 60 dpv. The average NAb titers of serum samples from the vaccine group were 64.67 and 76.38 at 30 and 60 dpv, respectively; these titers are significantly higher than the titers of corresponding samples from the PBS control group (Table 2; $P < 0.05$). At 188, 204, and 285 dpv, the NAb titers were negative for all serum samples from the vaccine group. Fish in the PBS control group had no detectable NAb titer at any postvaccination time point. These results suggest that the Montanide GEL 02 PR-adjuvanted IHN inactivated vaccine induced specific NAb in rainbow trout as early as 30 dpv, which could last at least until 60 dpv (Table 2).

**Viral titers in vaccinated fish after IHNV challenge.** Fish were challenged at 285 dpv, and whole liver, spleen, and kidney were collected and pooled from the dead fish or at 16 dpc from the fish that were still alive; serum samples were also collected at 16 dpc from the surviving fish. The virus loads in the tissues of the fish killed by IHNV were significantly higher than the virus loads in the tissues of fish that survived the IHNV challenge ($P < 0.05$; Fig. 6). Among the groups, the fish killed by IHNV in the PBS group had the highest virus titer ($10^{6.53}$ TCID$_{50}$/0.02 g); the virus titer in the tissue of

**TABLE 2** Duration of neutralizing antibody (NAb) response elicited by 50 $\mu$L dosage of Montanide GEL 02 PR inactivated vaccine (containing 45 $\mu$L inactivated IHNV)

| Days postvaccination (dpv) | No. of fish seroconverted/no. tested | NAb titers |
|---|---|---|
| 30 | 5/5 | 64.67 ± 6.93 |
| 60 | 5/5 | 76.28 ± 17.94 |
| 188 | 0/5 | <20 |
| 204 | 0/5 | <20 |
| 285 | 0/5 | <20 |

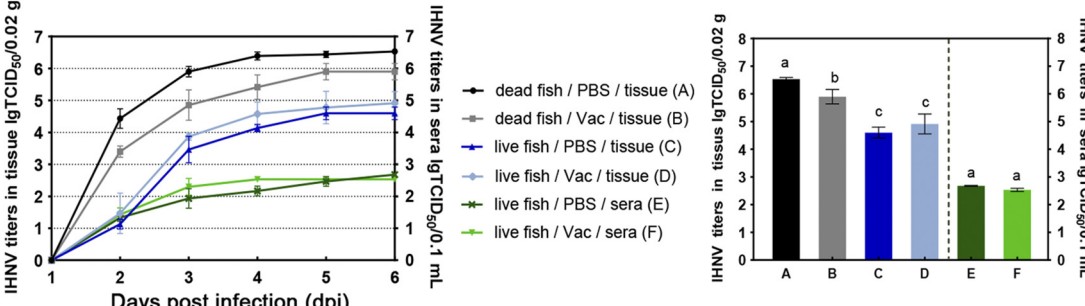

**FIG 6** Virus titers in the tissues and sera of vaccinated fish after IHNV challenge. Fish were challenged with IHNV stain LN15 at 285 dpv, and tissues (liver, spleen, and head kidney) were collected from the dead fish and from the surviving fish at 16 dpc. Serum samples were also collected from the surviving fish and titrated. Viral loads in the tissue and serum samples were statistically analyzed separately.

fish killed by IHNV in the immunized group was significantly lower ($10^{5.90}$ TCID$_{50}$/0.02 g). Neither the virus titers in the tissues ($10^{4.60}$ TCID$_{50}$/0.02 g and $10^{4.92}$ TCID$_{50}$/0.02 g, respectively) nor those in the sera ($10^{2.68}$ TCID$_{50}$/0.01 mL and $10^{2.53}$ TCID$_{50}$/0.01 mL, respectively) of the surviving fish in the PBS group and the immunization group were statistically different from one another (Fig. 6).

## DISCUSSION

IHN inactivated vaccines have been prepared mainly by using the inactivators BPL, formaldehyde, or binary ethylenimine (BEI) (7 to 9, 24). The costs of BPL and BEI are relatively high, and there are particular requirements for their preservation conditions. In comparison, formaldehyde is much cheaper and readily available and is the most commonly used inactivator in currently approved aquatic viral vaccines, including those for viruses, such as infectious pancreatic necrosis virus (IPNV) and infectious salmon anemia virus (ISAV), and those for bacteria, such as *Salmonella aeromonas* and *Flavobacterium column* (17). In previous studies, Anderson (7) and Tang (8) found that the immunity protection effect of the BPL-inactivated vaccine against IHN was higher than that of the formaldehyde- or BEI-inactivated vaccine against IHN. In contrast, we did not find a significant difference between the immune protective effects of the formaldehyde-inactivated vaccine against IHN and the BPL-inactivated vaccine against IHN in rainbow trout, although the immune protective effect of the formaldehyde-inactivated vaccine against IHN trended higher. Our results are similar to those of a previous study (9), in which an IHN inactivated vaccine developed with formaldehyde and one developed with BPL both had a good protective effect on rainbow trout, and the protective effect of the formaldehyde-inactivated vaccine was higher; together, these findings suggest that formaldehyde is superior to BPL for use in the preparation of an IHN vaccine for rainbow trout. Thus, owing to the advantages and wide application of formaldehyde, although no statistical significance was observed between the vaccines generated by the two inactivators tested in this study, formaldehyde was finally selected to prepare IHN inactivated vaccine.

To find a means of stabilizing the long-term protection induced by the formaldehyde-inactivated vaccine against IHN, adjuvants were screened in this study. We selected four kinds of adjuvants: a bidirectional oil adjuvant (GR208), an aluminum adjuvant (aluminum hydroxide gel), a non-mineral oil adjuvant (Montanide ISA 763A V), and a water-based adjuvant (Montanide GEL 02 PR). The aluminum adjuvant is the first adjuvant category approved by the U.S. Food and Drug Administration (FDA) for use in human vaccines (25). Presently, aluminum adjuvants are widely used in human and animal vaccines (26). Oil emulsion adjuvants are also one of the most commonly used adjuvants in human and animal vaccines. In the field of fish vaccinology, Montanide series oil emulsion adjuvants, such as Montanide ISA 761 VG and Montanide ISA 763A VG (Seppic, France), are the most widely used and have been proven to be effective in rainbow trout (27, 28) and turbot (29). However, in some cases, the use of oil adjuvants

in vaccines resulted in considerable side effects, such as intra-abdominal adhesions in Atlantic salmon (30) and Atlantic halibut (*Hippoglossus L.*) (31) and decreased appetite, bad growth rates (32, 33), and spinal deformity (33) in Atlantic salmon. In recent years, a number of studies revealed that the Montanide GEL 01 ST (Seppic, France), a novel water-based adjuvant, can effectively improve the protective effect of vaccines with few or no side effects in mice and pigs (34–36); however, there are no reports of this adjuvant being tested in fishery vaccines. The adjuvant Montanide GEL 02 PR is a new water-based adjuvant updated from the Montanide GEL 01 ST, and there are limited studies on it to date. In the present study, we found that the Montanide GEL 02 PR not only possessed promising immunity enhancement properties but also was highly safe in rainbow trout, without any side effects, such as the systemic histopathological damage or inhibition of growth (data not shown) previously observed in other adjuvant-vaccine combinations.

During the preparation of these adjuvanted IHNV inactivated vaccines, although the compatibility of adjuvants and inactivated IHNV was performed in accordance with the manufacturer's instructions and the methods described in a previous study (37), the prepared Montanide ISA 763A VG-adjuvanted IHN inactivated vaccine was found to be difficult to inject using a disposable sterile syringe (0.45 mm $\times$ 13.5 mm, Changchun Minjian Medical Instrument Co., LTD, China), which may have been caused by the size of the emulsion particles being relatively large in comparison with the size of the syringe needle. Therefore, the adjuvant Montanide ISA 763A VG was not included in this study. Among the three kinds of tested adjuvants, the Montanide GEL 02 PR was found to have the best performance, whereas the GR208 and aluminum hydroxide were ineffective. In previous studies, the GR208 was proven to be very efficient when combined with an inactivated vaccine against grass carp hemorrhagic disease (38), and aluminum hydroxide gel did not show a sufficient immune protection effect in yellow croaker (*Larimichthys crocea*) (37). These findings suggest that there is no single adjuvant that would be generally suitable for all types of vaccines or host species; thus, the evaluation of adjuvants is necessary for vaccine development.

Several kinds of adjuvants have been used to prepare IHN inactivated vaccines, and high immune protection in rainbow trout was obtained at early time points (7 to 9). Tang (8) found that an IHN inactivated vaccine adjuvanted with mineral oil (Sigma) had an RPS of 79.17% at 56 dpv, Anderson (7) illustrated that one adjuvanted with Freund's complete adjuvant (FCA) had an RPS of 70% at the same time point (56 dpv), and Gao (9) observed that one adjuvanted with Montanide ISA 206 adjuvant had RPSs ranging from 85% to 97.5% at 60 dpv. A recent study (11) showed that an attenuated IHN vaccine administered intranasally or intramuscularly provided an RPS of 40.21% or 25.78%, respectively, at 6 months postvaccination (mpv). Regrettably, no long-term immune protection of an IHN inactivated vaccine was tested in any previous studies. In the present study, we found that the protection efficacy of the IHN inactivated vaccine could be dramatically enhanced by its combination with an appropriate adjuvant. We determined the long-term protection of two dosages of the candidate vaccine, and the higher vaccine dosage provided complete protection (RPS of 100%) at 6.8 mpv, which is even better than the level of protection at a similar postvaccination time point provided by a previously reported DNA vaccine (RPS of 69%) (3) or the attenuated vaccine (RPS of 40.21% for intranasal administration and 25.78% for intramuscular administration) (11). Furthermore, our prepared adjuvanted IHN vaccine could maintain its high protection efficiency to 285 dpv (9.5 mpv) with an RPS of 60.79%. Kurath et al. (3) found that the IHN DNA vaccine could maintain its RPS of approximately 60% from 6 mpv to 2 years postvaccination. Further studies are required to determine if our vaccine can also maintain its protection effect for a similarly long period.

NAb is thought to play a crucial role in preventing the spread of virus particles throughout the body of an immunized subject, thereby preventing the subsequent progression of disease (39). In general, vaccines that can induce a high NAb titer in the immune target usually induce a better immune response (31). In this study, NAb was

detected at 1 to 2 mpv but dropped to undetectable levels at 6 mpv. Although no detectable NAb was observed at later time points after immunization, high immune protection rates were still obtained in this study; a similar phenomenon was also found for immunization with the well-known IHNV DNA vaccine at 6 mpv and 25 mpv (3). In a previous study of the IHN DNA vaccine in rainbow trout, Kurath et al. defined this phase as the long-term antiviral response (LAVR) in rainbow trout (3). The LAVR is characterized by reduced but still significant levels of protective immunity and a reduction of NAb titers and seroprevalence. In the present study, our candidate vaccine could induce significant immune protection in rainbow trout against an IHNV challenge at later time points, at which no detectable NAb was found in vaccinated fish. In another study, Ma et al. (11) found that the antibody levels (as determined by ELISA) at 6 mpv in rainbow trout vaccinated with an attenuated IHN vaccine were not significantly different from those in unvaccinated fish, but the difference in immune protection between these groups was significant. These findings further confirm that the LAVR is a common mechanism of long-term protection elicited by vaccination in rainbow trout. We also quantified the virus in surviving fish after the IHNV challenge, and the virus was found in both the vaccination group and the negative control group. The level of virus in the surviving fish was lower than that in the fish killed by the IHNV challenge, and this level in vaccinated fish was lower than that in negative control (mock-vaccinated) fish. These findings indicate that our candidate IHN vaccine provided protection by reducing the virus load, not by clearing the virus, during the LAVR. Kurath et al. proposed two possible explanations regarding the immune mechanism of the LAVR. One is that some level of antibody below our limit of detection may be present at or near the threshold titer needed for protection, and the other is that there might be other factors, such as cellular immunity, that may be even more important than humoral factors (3); this topic requires further research.

## MATERIALS AND METHODS

**Fish.** Specific-pathogen-free rainbow trout were obtained from the Agrimarine Company, Benxi, China. The immunized fish were maintained in spring water flow-through tanks (diameter: 2 m; water depth: 1 m) inside of a company hatchery. At different time points after immunization, fish were transported back to our laboratory 7 days prior to challenge experiments. Challenged fish were maintained in separated tanks (50 cm × 50 cm × 60 cm) supplied with circulating water at a temperature of 13 ± 1°C. In the challenge experiments, each treatment subgroup contained three replicates. The exact sizes of the experimental fish used in this study are described in the corresponding sections.

**Viruses and cell lines.** The IHNV strain LN15 (2), isolated from diseased rainbow trout and stored in our laboratory, was used for the preparation of the IHN inactivated vaccine and viral challenges in this study. The $TCID_{50}$ of the IHNV strain LN15 is $1 \times 10^{-7}/0.1$ mL, and the IHNV with this titer was used for all challenge experiments unless otherwise specified. The *Epithelioma papulosum cyprini* (EPC) cell line, derived from fathead minnow (*Pimephales promelas*; ATCC CRL-2872), generously provided by Prof. Zeng from Yangtze River Fisheries Research Institute, Chinese Academy of Fishery Sciences, was maintained at 25°C in minimum essential medium (MEM) supplemented with 10% fetal bovine serum (FBS). IHNV was propagated in EPC cells in MEM supplemented with 2% FBS at 15°C (14).

**Selection of IHNV inactivator.** Freshly amplified IHNV was inactivated with formaldehyde or beta-propiolactone (BPL), and the immune protection of these prepared inactivated vaccines was compared by viral challenge tests in two sizes of rainbow trout (average weight of 6 ± 2 g or 70 ± 8 g). The inactivation protocols, established in our laboratory, were as follows: formaldehyde or BPL was mixed with live IHNV at a final concentration of 5 mmol/L or 3.0 mmol/L, respectively, the resulting mixture was agitated at 100 rpm for 24 h at room temperature, and the inactivation reaction was terminated by neutralization with sodium bisulfite with a final concentration of 0.01% for formaldehyde or with sodium thiosulfate with a final concentration of 20 mmol/L for BPL.

Juvenile rainbow trout (weighing 6 ± 2 g) were randomly divided into three groups of 60 fish each, and each group was divided into three parallel subgroups. Fish were intraperitoneally injected with 50 $\mu$L of formaldehyde-inactivated vaccine or BPL-inactivated vaccine (immunization groups) or with the same volume of phosphate-buffered saline (PBS) (negative control group). To exclude the influence of the vaccine inactivator, fish were injected with equal amounts of formaldehyde or BPL contained in the same volume of PBS. At 60 days postvaccination (dpv), the immunized fish were anesthetized and challenged with 50 $\mu$L of a suspension of IHNV strain LN15 (100 $TCID_{50}$). The cumulative percentage mortality (CPM) in each group was recorded daily for 21 days. The relative percent survival (RPS) was calculated using the following formula: RPS = (1 – average CPM of immunization group/average CPM of PBS control group) × 100%.

The protective efficacies of these two inactivated vaccines in adult rainbow trout (weighing 70 ± 8 g) were evaluated by the same method, but the dosages of the vaccines and challenge virus were both adjusted to 100 $\mu$L per fish.

**TABLE 3** Information of adjuvants used in this study

| Trade names (article no.) | Types | Sources | Ratio (adjuvant: inactivated virus, v/v) | Recommended methods for mixture |
|---|---|---|---|---|
| GR208 | Bidirectional oil adjuvant | Guorui Biotechnology Co., Ltd., Changsha, China | 1: 1 | Low shear force (2,000 to 5,000 rpm) for 5 minutes |
| Aluminum hydroxide gel (BG1119) | Aluminum adjuvant | Zhonghui Hecai Biomedical Technology Co., Ltd., Shanxi, China | 4: 1 | Manual rock up and down |
| Montanide ISA 763A V (36084X) | Non-mineral oil adjuvant | Seppic, Paris, France | 7: 3 | High shear force (10,000 to 12,000 rpm) for 5 minutes |
| Montanide GEL 02 PR (36017Z) | Water-based adjuvant | Seppic, Paris, France | 1: 9 | Manual rock up and down |

**Screening of adjuvants.** Four commercial adjuvants, GR208, aluminum hydroxide gel, Montanide 763A VG, and Montanide GEL 02 PR, were separately mixed with IHN inactivated vaccines to prepare adjuvanted inactivated vaccines. The methods used to combine the vaccines and adjuvants are shown in Table 3. Before it was combined with the IHN vaccine, each adjuvant was sterilized at 116℃ for 20 min in a high-temperature and -pressure container. The sterility of the prepared adjuvanted inactivated vaccine for IHN was verified in accordance with the standards of "Chinese Veterinary Pharmacopoeia" before each animal experiment.

Juvenile rainbow trout (weighing $7 \pm 1.5$ g) were intraperitoneally injected with an adjuvanted inactivated vaccine for IHN, adjuvant alone, or PBS at a dosage of 50 $\mu$L. The immunized fish were maintained in tanks inside of the hatchery on the fish farm. At 60 dpv, the vaccinated fish that had been transferred in advance were maintained in separate tanks in the laboratory for challenge experiments and neutralizing antibody (NAb) titer evaluation. The fish ($n = 60$) from each treatment were intraperitoneally injected with 50 $\mu$L of IHNV strain LN15 (100 $TCID_{50}$). The CPM was recorded daily for 21 days. Blood samples of rainbow trout ($n = 5$) collected from the caudal veins of the fish were used for the determination of NAb titers as described in section NAb Titer Determination.

**NAb titer determination.** Blood samples were collected from the caudal veins of fish ($n = 5$) at specified time points after immunization. The blood samples were maintained overnight at 4℃ and then centrifuged at 500 $\times$ g for 10 min to isolate the serum. The serum samples were continuously 2-fold diluted (from 1:20 to 1:320) with MEM supplemented with 2% FBS and then mixed with an IHNV suspension (100 $TCID_{50}$) in a ratio of 1:1. The serum–virus mixture was incubated in a $CO_2$ incubator at 15℃ for 1 h and then used to inoculate a 96-well monolayer of EPC cells. The 96-well cell plates were incubated for 1 h in a $CO_2$ incubator at 15℃, after which the cell culture medium containing serum and IHNV was discarded and replaced with 100 $\mu$L of MEM supplemented with 2% FBS per well. The EPC cell culture plate was statically cultured at 15℃ for 7 d, and the number of infected and noninfected cell wells was recorded. The serum antibody titers were determined as the reciprocal of the serum dilution that reduced the viral infectivity by approximately 50% relative to the virus control. Samples with an NAb titer of $\geq$20 were considered positive, whereas those with an NAb titer of <20 were considered negative, as in a previous study (40).

**The optimal dosage of adjuvanted inactivated vaccine against IHN.** After identifying the adjuvant with the best immune-enhancing effect in our screening experiments described above, we conducted experiments to compare the protective ability of different dosages of the optimal adjuvanted inactivated vaccine in rainbow trout. Healthy rainbow trout (weighing $6 \pm 2$ g) were randomly divided into four groups of 45 fish each, and each group was further divided into three parallel subgroups. Three of the groups (groups 1 to 3) were respectively immunized with three different dosages of Montanide GEL 02 PR-adjuvanted inactivated vaccine (Table 4). The last group (group 4) was treated with PBS and used as the negative control group. At 30 dpv, the fish in each group were intraperitoneally injected with IHNV strain LN15 (100 $TCID_{50}$) at a dosage of 50 $\mu$L per fish. The CPM and RPS were compared among groups to define the protective ability of the Montanide GEL 02 PR-adjuvanted IHN inactivated vaccine at different dosages.

**Duration of immune protection.** To evaluate the long-term immune protection, juvenile rainbow trout ($8 \pm 2.4$ g) were vaccinated with 50 $\mu$L of Montanide GEL 02 PR-adjuvanted IHN inactivated vaccine, containing 9 $\mu$L or 45 $\mu$L of inactivated IHNV. Fish mock-vaccinated with PBS were used as the negative control. At 30, 60, 188, 204, and 285 dpv, fish ($n = 60$) previously immunized with Montanide GEL

**TABLE 4** Dosages of Montanide GEL 02 PR inactivated vaccine

| Groups | Treatments | Ratio (inactivated virus: PBS: adjuvant, v/v/v) | Total immunization dosage/contained inactivated virus vol ($\mu$L) |
|---|---|---|---|
| 1 | Montanide GEL 02 PR vaccine | 9: 36: 5 | 50/9 |
| 2 | Montanide GEL 02 PR vaccine | 45: 0: 5 | 50/45 |
| 3 | Montanide GEL 02 PR vaccine | 90: 0: 10 | 100/90 |
| 4 | PBS control | 0: 100: 0 | 100/0 |

**TABLE 5** Dosage of IHNV used in challenge experiments at different time points after immunization

| Treatments | Challenge time points (dpv) | The mean wt (g) | Concn of IHNV suspension (TCID$_{50}$) | Challenge dosage of IHNV ($\mu$L) |
|---|---|---|---|---|
| 50 $\mu$L Montanide GEL 02 PR vaccine (containing 45 $\mu$L inactivated IHNV) or equal vol PBS | 30 | 14 ± 1.3 | 10$^2$ | 50 |
| | 60 | 23 ± 2.1 | 10$^2$ | 50 |
| | 188 | 48 ± 2.9 | 10$^4$ | 100 |
| | 204 | 53 ± 4.3 | 10$^6$ | 100 |
| | 285 | 64 ± 6.5 | 10$^6$ | 100 |
| 50 $\mu$L Montanide GEL 02 PR vaccine (containing 9 $\mu$L inactivated IHNV) or equal vol PBS | 28 | 13 ± 2.1 | 10$^2$ | 50 |
| | 56 | 22 ± 2.8 | 10$^2$ | 50 |
| | 70 | 26 ± 3.9 | 10$^3$ | 50 |
| | 144 | 45 ± 5.8 | 10$^6$ | 100 |
| | 223 | 57 ± 6.7 | 10$^6$ | 100 |

02 PR-vaccine containing 45 $\mu$L of inactivated virus were challenged with IHNV strain LN15. The CPM and RPS at 21 days postchallenge were measured. The NAb titers ($n = 5$) were measured at each time point. At 28, 56, 70, 144, and 223 dpv, fish previously immunized with Montanide GEL 02 PR-vaccine containing 9 $\mu$L of inactivated virus were challenged with IHNV strain LN15. Negative control fish were challenged along with the vaccinated fish at each time point, and the virus dosages used in each challenge experiment are shown in Table 5.

**Post-challenge viral titers in fish.** After the viral challenge at 285 dpv, we also detected the virus titers in the tissues and sera of fish from the immunization group and the PBS control group. We collected the dead fish ($n = 5$) and the live fish ($n = 5$) that survived for 16 days postchallenge (dpc) (when mortality ceased). The whole liver, spleen, and anterior kidney tissues were removed and pooled, thoroughly homogenated, and diluted with PBS at a ratio of 1:5 (g:mL). The pooled tissue suspension was centrifuged at 1200 $\times$ $g$ for 5 min to harvest the supernatant. Blood samples were collected from the caudal veins of surviving fish, stored overnight at 4°C, and then centrifuged at 500 $\times$ $g$ for 10 min to obtain the serum. These tissue and serum samples were subjected to continuous 10-fold dilutions and then used to inoculate EPC cells. The daily viral titers were measured in accordance with the Reed-Muench method (41). Each sample was tested in three independent experiments.

**Statistical analysis.** Statistical analysis was performed on the results of each test. The statistical significance was determined by conducting a one-way analysis of variance (ANOVA) or $t$ test. $P$ values of $< 0.05$ were considered to indicate a statistically significant difference. The data analysis was conducted using GraphPad Prism 8.0.1.

**Ethical approval.** All experiments were conducted based on local government regulations, and fish experiments were approved by the Institutional Animal Care and Use Committee of Heilongjiang River Fisheries Research Institute, Chinese Academy of Fishery Sciences.

## ACKNOWLEDGMENTS

We declare that we have no conflicts of interest.

This work was supported by the National Key Research and Development Program of China (2019YFE0115500), the National Natural Science Foundation of China (32002437), and the Central Public-Interest Scientific Institution Basal Research Fund, CAFS (2020TD43).

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
