## [Reviewer comments · Microbiology Spectrum]

Microbiology Spectrum

Long-term protection elicited by an inactivated vaccine supplemented with a water-based adjuvant against infectious hematopoietic necrosis virus in rainbow trout (*Oncorhynchus mykiss*)

Yujie Lin, Guangming Ren, Jingzhuang Zhao, Yizhi Shao, Baoquan He, Xin Tang, Ou Sha, Wenwen Zhao, Qi Liu, Liming Xu, and Tongyan Lu

Corresponding Author(s): Liming Xu, Heilongjiang Fisheries Research Institute, Chinese Academy of Fishery Sciences

Review Timeline:

Submission Date:	August 18, 2022
Editorial Decision:	September 1, 2022
Revision Received:	October 10, 2022
Accepted:	October 21, 2022

Editor: Daniela Rajao

Reviewer(s): Disclosure of reviewer identity is with reference to reviewer comments included in decision letter(s). The following individuals involved in review of your submission have agreed to reveal their identity: Lingbing Zeng (Reviewer #1); Weiwei Zeng (Reviewer #2)

Transaction Report:

DOI: <https://doi.org/10.1128/spectrum.03245-22>

September 1, 2022

Dr. Liming Xu
Heilongjiang Fisheries Research Institute, Chinese Academy of Fishery Sciences
Harbin
China

Re: Spectrum03245-22 (Long-term protection elicited by an adjuvant- inactivated vaccine against infectious hematopoietic necrosis virus in rainbow trout (*Oncorhynchus mykiss*))

Dear Dr. Liming Xu:

Thank you for submitting your manuscript to Microbiology Spectrum. Your manuscript has now been reviewed by experts in the field. Both reviewers agree that the manuscript is important and the methods are sound. However, some revisions are needed, particularly on the organization of the manuscript and inclusion of important descriptive information.

Link Not Available

Sincerely,

Daniela Rajao

Journals Department
Reviewer comments:

Reviewer #1 (Comments for the Author):

Salmon and trout aquaculture represents one of the most active industry in Fisheries, providing huge amount of fish meal protein for human beings. However, these species of fish have a severe disease caused by IHNV. It is of significant importance to investigate the approached to control this fish viral disease, including vaccine innovation. In this manuscript, the author fully investigated a adjuvant-inactivated vaccine for IHN and revealed a long-term protection in this fish. The chose of research subject is right, the methods sed are reasonable and the results are reliable. However, this manuscript should have a major

revision as follows:

1. Major comments:

- 1) Re-organization of the manuscript. Different aspects related to the preparation of adjuvant-inactivated vaccine are investigated in this study, including optimal inactivator, optimal adjuvant, optimal dosage of vaccine, viral titers in tissue and serum after challenge, NAb titration after challenge, etc. to a great extent, the author should shorten the manuscript and focus on the duration of the immune response and the protection elicited by the vaccine;
- 2) Please ask for help from an English-native speaker to polish the language of this manuscript.

2. Minor comments

- 1) Line 35: what does "were useless" mean? please revise this description;
- 2) Line 38: May the "bigger dosage" be corrected to "higher dosage"?
- 3) It is suggested the author presents a very brief introduction of the salmon and trout industry in the part of Introduction before introducing the state of the IHNV.
- 4) Line 98-99: "The unsatisfactory....face" please revised to "The unsatisfactoryis one of the most common problems of inactivated vaccine";
- 5) Line 108: " are relative single, mostly oil adjuvants" should be revised to " are poor, most of them are oil adjuvant";
- 6) Line 127: "virus strains and cell lines" should be "Viruses and cell lines";
- 7) Line 132: "donated" should be "generously presented by...";
- 8) Line 136-139: delete this sentence;
- 9) Line 162: do not use small or large to describe the fish size used in research, please present the detailed size in average centimeters or average weight.
- 10) Results: please carefully select those results to be presented here, i.e, those "optimal inactivator, optimal adjuvant, optimal dosage" should be included in this manuscript? moreover, what the purposes are : Line 435 " viral titers in vaccinated fish after challenge", Line 452 "NAb titers after challenge"?
- 11) Line 460 3.7 what the purpose is? "Pathogenicity analysis of IHNV retained in the challenged fish";
- 12) Fig 8: please adjust the figures, they are out of shape to some extent.

Reviewer #2 (Comments for the Author):

This article studied an inactivated vaccine that could elicit long-term protection against IHNV in rainbow trout. The experimental design of the article is reasonable but there are still some concerns that need to be corrected before publication

1. The title is "adjuvant-inactivated vaccine" is that means the vaccine was inactivated by adjuvant? This sentence is ambiguous. Such as "2.11. Safety evaluation of MontanideTM GEL 02 PR inactivated vaccines", this means the vaccine is inactivated by MontanideTM GEL 02 PR? Please change to a more rigorous statement.
 2. For all virus challenge, the author only defined virus dose in the selection of IHNV inactivator part, but other experiment the author only provided virus volume. Is these use the same virus? If the concentrations of virus are same, the author need to define it somewhere.
 3. Line 83, Please define non-GMO.
 4. Please provide the detail information of four commercial adjuvants product to allow others to repeat this experiment.
 5. Please unify the format of "Please unify the format."i.p." or "i.p".
 - 6.3.2.1. Please define the group of RPS: 30.51%, 78.68%, 36.4%, and 30.33%
- Unify the reference format, Reference 1 "Mol Phylogenet Evol", 11 "J Fish Dis", 14 "J Virol", 36 "Dis Aquat Org"

Staff Comments:

Preparing Revision Guidelines

For complete guidelines on revision requirements, please see the journal Submission and Review Process requirements at <https://journals.asm.org/journal/Spectrum/submission-review-process>. **Submissions of a paper that does not conform to**

Microbiology Spectrum guidelines will delay acceptance of your manuscript. "

Please return the manuscript within 60 days; if you cannot complete the modification within this time period, please contact me. If you do not wish to modify the manuscript and prefer to submit it to another journal, please notify me of your decision immediately so that the manuscript may be formally withdrawn from consideration by Microbiology Spectrum.

Reviewers' comments

Reviewer #1: Salmon and trout aquaculture represents one of the most active industry in Fisheries, providing huge amount of fish meal protein for human beings. However, these species of fish have a severe disease caused by IHNV. It is of significant importance to investigate the approached to control this fish viral disease, including vaccine innovation. In this manuscript, the author fully investigated an adjuvant-inactivated vaccine for IHN and revealed a long-term protection in this fish. The choice of research subject is right, the methods sed are reasonable and the results are reliable. However, this manuscript should have a major revision as follows:

1. Major comments

1) Re-organization of the manuscript. Different aspects related to the preparation of adjuvant-inactivated vaccine are investigated in this study, including optimal inactivator, optimal adjuvant, optimal dosage of vaccine, viral titers in tissue and serum after challenge, NAb titration after challenge, etc. to a great extent, the author should shorten the manuscript and focus on the duration of the immune response and the protection elicited by the vaccine;

Response: Thank you very much for your suggestion. We are completely agreed with your professional opinions. The main aim of the study was to prolong the protection duration of an inactivated vaccine against IHNV by introducing an appropriate adjuvant. All the experiments should be focus on this theme. So, "Pathogenicity tests for IHNV retained in the challenged fish" and "Safety evaluation of the inactivated vaccines", and "NAb Titration after challenge" were deleted from the revised manuscript.

2) Please ask for help from an English-native speaker to polish the language of this manuscript.

Response: Thank you very much for your suggestions. The manuscript has been revised by a native English speaker.

2. Minor comments

1) Line 35: what does "were useless" mean? Please revise this description;

Response: We wanted to express the enhancement effect of other adjuvants is not obtained. We have revised this description at line 35-36 in the revised manuscript (Clearcopy).

2) Line 38: May the "bigger dosage" be corrected to "higher dosage"?

Response: We have replaced the “bigger dosage” with “higher dosage” at line 39 in the revised manuscript (Clearcopy).

3) It is suggested the author presents a very brief introduction of the salmon and trout industry in the part of Introduction before introducing the state of the IHNV.

Response: Thank you very much for your advice, which will be much of significance. We have made the corresponding content supplement at the beginning of the introduction section at line 62-66 in the revised manuscript (Clearcopy).

4)Line 98-99: "The unsatisfactory....face" please revise to "The unsatisfactoryis one of the most common problems of inactivated vaccine";

Response: The description has been revised at line 99-100 in the revised manuscript (Clearcopy).

5)Line 108: " are relative single, mostly oil adjuvants" should be revised to " are poor, most of them are oil adjuvant";

Response: Thank you for your suggestion. We have revised this sentence at line 108 in the revised manuscript (Clearcopy).

6) Line 127: "virus strains and cell lines "should be "Viruses and cell lines";

Response: Thank you very much for your correction. We have already corrected "virus strains and cell lines" to "Viruses and cell lines" at line 126 in the revised manuscript (Clearcopy).

7) Line 132: "donated" should be "generously presented by...";

Response: We have replaced the "donated" with "generously presented by..." at line 132 in the revised manuscript (Clearcopy).

8) Line 136-139: delete this sentence;

Response: Thank you very much for your correction. We have deleted the two unnecessary elaboration at line 136-139 in the initial manuscript.

9) Line 162: do not use small or large to describe the fish size used in research, please

present the detailed size in average centimeters or average weight.

Response: We have delated the misexpression of describing the experimental fish small and large, instead we used the average weight of the experimental fish in the revised manuscript (Clearcopy).

10) Results: please carefully select those results to be presented here, i.e, those "optimal inactivator, optimal adjuvant, optimal dosage" should be included in this manuscript? moreover, what the purposes are : Line 435 "viral titers in vaccinated fish after challenge", Line 452 "NAb titers after challenge"?

Response: Thank you for your suggestion. It really was not appropriate to used so many the "optimal" in such a simple study. We have revised this kind of description at line 137 and 162 in the revised manuscript (Clearcopy).

We did not find detectable NABs at late time points after vaccination but obvious protection was observed. We would like to reveal the viral status in challenged fish. So, we determined the viral titers in fish after challenge. We found that vaccinated fish had lower viral titers than mock vaccinated fish., which indicated that the inactivated vaccine provided protection to rainbow trout by reducing virus loads not by clearance of virus at late time point post vaccination. This experiment was meaningful for understanding the whole study, we keep it at line 228-240 and 363-375 in the revised manuscript (Clearcopy).

NAb was titrated in challenged fish based on the assumption that the NAb might be dramatically induced again by memory B cells when exposure to virus. But we did not find the detectable NABs in the challenged fish. So, we did not include this part of experiments in the revised manuscript (Clearcopy).

11) Line 460 3.7 what the purpose is? "Pathogenicity analysis of IHNV retained in the challenged fish";

Response: We found that viral titers in survived fish were high, which made us to think if the remained virus in fish possess the pathogenicity or not. So, we isolated the virus and test it through challenging rainbow trout. It turned out that the isolated virus was still very virulent to rainbow trout. This part is not so relevant to this study, so we deleted this part at line 261-275 and 460-476 in the initial manuscript.

12) Fig 8: please adjust the figures, they are out of shape to some extent.1

Response: Thank you very much for your advice. In this revision, we have carefully modified each picture according to the guidelines for author.

Reviewer #2: This article studied an inactivated vaccine that could elicit long-term protection against IHNV in rainbow trout. The experimental design of the article is reasonable but there are still some concerns that need to be corrected before publication.

1. The title is "adjuvant-inactivated vaccine" is that means the vaccine was inactivated by adjuvant? This sentence is ambiguous. Such as "2.11. Safety evaluation of MontanideTM GEL 02 PR inactivated vaccines", this means the vaccine is inactivated by MontanideTM GEL 02 PR? Please change to a more rigorous statement.

Response: Thank you for your advice. Now we realize that the expression of "adjuvant-inactivated vaccine" is not precise enough and easily lead to misunderstanding. We have used "inactivated vaccine supplemented with an adjuvant" to take place of the "adjuvant-inactivated vaccine" at line 2-3 in the revised manuscript (Clearcopy).

2. For all virus challenge, the author only defined virus dose in the selection of IHNV inactivator part, but other experiment the author only provided virus volume. Is these use the same virus? If the concentrations of virus are same, the author need to define it somewhere.

Response: The experimental fish had inconsistent sizes in these experiments, so that the virus doses used in the challenge were different. We have added the detailed information in each individual experiment in the revised manuscript (Clearcopy).

3. Line 83, Please define non-GMO.

Response: GMO is the abbreviation of genetically modified organism. We have explained it in the revised at line 88-89 in the revised manuscript (Clearcopy).

4. Please provide the detail information of four commercial adjuvants product to allow others to repeat this experiment.

Response: Thank you for your advice, and we have added description of the adjuvants used in this study in Table. We briefly give some basic information about the four adjuvants, including types, commercial sources, recommended optimal ratio

of adjuvant to vaccine and recommended methods for mixture of adjuvants and vaccine.

5. Please unify the format of "Please unify the format. "i.p." or "i.p".

Response: Thank you for your correction which helped us a lot. We have used "i.p" to take place of "i.p".

6.3.2.1. Please define the group of RPS: 30.51%, 78.68%, 36.4%, and 30.33%

Response: We have revised the description to "The RPSs of the groups vaccinated with the non-adjuvanted vaccine or the vaccines supplemented with aluminum hydroxide gel, Montanide™ GEL 02 PR , or GR208 were 30.33%, 30.51%, 78.68%, and 36.4%, respectively" at line 272-275 in the revised manuscript (Clearcopy).

Unify the reference format, Reference 1 "Mol Phylogenet Evol",11 "J Fish Dis",14 "J Virol", 36 "Dis Aquat Org"

Response: we have checked all the references and uniformed them in the revised manuscript (Clearcopy).

October 21, 2022

Dr. Liming Xu
Heilongjiang Fisheries Research Institute, Chinese Academy of Fishery Sciences
Harbin
China

Re: Spectrum03245-22R1 (Long-term protection elicited by an inactivated vaccine supplemented with a water-based adjuvant against infectious hematopoietic necrosis virus in rainbow trout (*Oncorhynchus mykiss*))

Dear Dr. Liming Xu:

Your manuscript has been accepted, and I am forwarding it to the ASM Journals Department for publication. You will be notified when your proofs are ready to be viewed.

Sincerely,

Daniela Rajao
Editor, Microbiology Spectrum
